# Challenges in managing HIV and non-communicable diseases and health workers' perception regarding integrated management of non-communicable diseases during routine HIV care in South Central Uganda: A qualitative study

**Asani Kasango**[1]*, **Alex Daama**[1,2], **Lilian Negesa**[1]

1 Department of Epidemiology and Clinical Research, Rakai Health Sciences Program, Kalisizo, Kyotera, Uganda, 2 Department of Science and Grants, African Medical and Behavioral Sciences Organization, Nansana, Wakiso, Uganda

* hassankasango25@gmail.com

## Abstract

### Background

Non-communicable diseases are highly prevalent among adults living with HIV, emphasizing the need for comprehensive healthcare strategies. However, a dearth of knowledge exists regarding the health systems challenges in managing HIV and non-communicable diseases and the perception of health workers regarding the integrated management of non-communicable diseases during routine HIV care in rural Ugandan settings. This study aims to bridge this knowledge gap by exploring the health system challenges in managing HIV and non-communicable diseases and health workers' perception regarding the integration of non-communicable diseases in routine HIV care in South Central Uganda.

### Methods

In this qualitative study, we collected data from 20 purposively selected key informants from Kalisizo Hospital and Rakai Hospital in South Central Uganda. Data were collected from 15th December 2020 and 14th January 2021. Data were analyzed using a thematic content approach with the help of NVivo 11.

### Results

Of the 20 health workers, 13 were females. In terms of work duration, 9 had worked with people living with HIV for 11–15 years and 9 were nurses. The challenges in managing HIV and non-communicable diseases included difficulty managing adverse events, heavy workload, inadequate communication from specialists to lower cadre health workers, limited financial and human resources, unsupportive clinical guidelines that do not incorporate non-communicable disease management in HIV care and treatment, and inadequate knowledge

**Data Availability Statement:** All relevant data underlying these findings is available on Figshare. https://doi.org/10.6084/m9.figshare.25262521.v1

**Funding:** The author(s) received no specific funding for this work.

**Competing interests:** The authors have declared that no competing interests exist.

and skills required to manage non-communicable diseases appropriately. Health workers suggested integrating non-communicable disease management into routine HIV care and suggested the need for training before this integration.

## Conclusion

The integration of non-communicable disease management into routine HIV care presents a promising avenue for easing the burden on health workers handling these conditions. However, achieving successful integration requires not only the training of health workers but also ensuring the availability of sufficient human and financial resources.

## Introduction

The global impact of the HIV pandemic has been profound, affecting over 70 million people, and resulting in more than 30 million deaths attributed to acquired immunodeficiency syndrome (AIDS) since the 1980s [1]. Notably, sub-Saharan Africa bears a significant burden, contributing to 90% of all people living with HIV (PLHIV) [2–4]. There has been successful implementation of HIV prevention, care, and treatment programs in low-income and middle-income countries (LMICs) since 2004 [5–7], resulting in higher survival rates of PLHIV [8–10], aligning with the progress observed in industrialized nations [11, 12].

As the world strives to achieve the Joint United Nations Programme on HIV/AIDS (UNAIDS)'s ambitious 95–95–95 goals (95% of people with HIV diagnosed, 95% on antiretroviral therapy (ART), and 95% virally suppressed by 2025) [13, 14], a shift in HIV prevalence becomes apparent. The decline in AIDS-related opportunistic infections accompanies a surge in the prevalence of non-communicable diseases (NCDs) [15–18]. Notably, in low- and middle-income countries (LMICs), PLHIV experiences a higher prevalence of NCDs, ranging from 29% to 44%, compared to the general population's rates of 15% to 25% [19, 20].

Despite the progress in ART accessibility, the interplay of factors such as toxicity of antiretrovirals (ARVs), microbial translocation, and persistent inflammation [21] in addition to the traditional risk factors for NCDs including overweight and obesity [22] contribute to NCD burden among PLHIV. Sub-Saharan Africa, in particular, grapples with heightened psychosocial stressors, poverty-linked stress, and suboptimal adherence to ART, potentially exacerbating the risk and impact of NCDs [23–25]. Despite improved access to efficacious ARVs leading to reduced mortality rates among PLHIV in Sub-Saharan Africa, the co-occurrence of HIV and NCDs is expected to rise [26–28]. Consequently, optimizing NCD care for PLHIV is becoming an increasingly critical global priority, particularly in sub-Saharan Africa, where the dual burden of NCDs and HIV is most pronounced [29].

Worldwide, NCDs contribute to over 36 million deaths annually, with 80% of these fatalities concentrated in resource-limited nations [30, 31]. The Global Burden of Disease 2020 underscores the escalating impact of NCDs on global health, particularly in resource-limited regions, including sub-Saharan Africa [32]. Cardiovascular diseases, cancer, chronic respiratory conditions, and metabolic disorders like diabetes account for a substantial portion of documented NCD-related deaths and chronic illnesses [33–35].

Beyond the individual health implications, NCDs place a huge burden on the healthcare system as they increase healthcare costs [36, 37] and overburden the health workers with the need to manage NCD independent of HIV. Furthermore, they require a lot of consultations and unplanned hospital admissions [38–40].

In Uganda, where approximately 6% of adults live with HIV [41], 20% of them contend with NCDs [42]. However, adherence to NCD treatment among this population remains disconcertingly low, highlighting a critical gap in care [43, 44]. Recognizing the potential benefits of integrating NCD management into routine HIV care, such as improvement in treatment adherence [45], this study seeks to explore the health system challenges associated with managing both HIV and NCDs among PLHIV in South Central Uganda. Additionally, it aims to unveil the perspectives of health workers on the integration of NCDs into routine care, addressing a significant knowledge gap in the context of sub-Saharan African settings, particularly Uganda.

## Methods

### Study design

We conducted a qualitative exploratory study utilizing key informant interviews with health workers.

### Setting and population

The research was conducted at Kalisizo Hospital and Rakai Hospital, which are Ministry of Health (MoH) facilities. Kalisizo Hospital is located in Kalisizo town, Kyotera district. The Hospital is approximately 30 km from Masaka Regional Referral Hospital. The Hospital offers a comprehensive range of services, encompassing outpatient and inpatient care, maternal and reproductive health, child health, nutritional support, HIV, and NCDs screening and management, as well as general healthcare. Serving as the primary hospital for the Kyotera district, it also acts as the referral center for lower-level health facilities within the district. The hospital's catchment population is estimated at around 70,000 individuals [46, 47], with a bed capacity of 120 beds. Over 87% of the population served live in rural areas that majorly depend on agriculture, trading, and fishing. HIV prevalence in Kyotera district ranges between 14% and 42% [48].

Rakai Hospital is an MOH Hospital located in the town of Rakai, Rakai district, about 61 km southwest of Masaka Regional Referral Hospital. It is the Main and only Hospital in the district, and it provides in-patient and outpatient services including maternal, reproductive, child health, nutrition, HIV, NCD screening and management, and general Healthcare. It is a predominantly rural facility with a bed capacity of 100 beds. Included in the study were health workers with a minimum of one year of experience in managing PLHIV who were employed at either Kalisizo Hospital or Rakai Hospital and consented to participate in the study.

### Sample size and recruitment

The sample size for the study was guided by saturation principle and 20 key informant interviews were done. Of the key informants, 12 were selected from Kalisizo Hospital and 8 were selected from Rakai Hospital. Purposive sampling was used to select key informants. This approach allowed for the targeted inclusion of health workers with specific and pertinent experiences, regarding HIV and NCD management at both hospitals.

### Data collection

Data were collected from 15th December 2020 and 14th January 2021 using a key informant interview guide with standardized open-ended questions. The tool contained questions about the common NCDs among PLHIV, questions on how they are managed, the challenges in their management, what could improve their management, and what they thought about the integrated management of NCDs in routine HIV care. With the key informant's permission,

field notes were taken and supplemented with audio recordings. All the key informant interviews were conducted in English and lasted 40 to 60 minutes.

## Data analysis

Responses were entered in Microsoft Excel 2010 by the corresponding author (AK) and quality was checked by (AD). Thematic and content analysis were conducted using NVivo software version 11 [49]. All significant statements from the interviews underwent open coding, eschewing pre-set codes. Subsequently, clusters of codes were identified and organized into themes. These themes were further circumscribed into thematic categories, forming analogical groups that were classified and aggregated based on the perceived meanings for the study participants. Our data reduction process unfolded sequentially, progressing from data familiarization to coding, and from theme generation to the definition and naming of these themes. The related themes were systematically categorized, leading to the creation of a comprehensive codebook. This book underwent a rigorous review and evaluation process involving all authors for verification. The findings were summarized as themes and frequencies, expressed as percentages. The interviews were analyzed and coded individually, with scrutiny of transcripts and field notes. This analytical process continued until data saturation was reached, signifying that we had gathered sufficient elements to meet the study's objectives. Quantitatively, descriptive analysis was done in Excel. In the final stages of analysis, the thematic categories and respective themes were examined and discussed, reflecting our commitment to adopting a comprehensive and interpretive approach to the results.

## Ethical considerations

Approval was secured from the Makerere University School of Public Health Higher Degrees Research and Ethics Committee (HDREC) before commencing data collection. Additionally, administrative clearances were obtained from the Kyotera and Rakai District Health Officer (DHO) and the Medical Superintendents of Kalisizo Hospital and Rakai Hospital before conducting key informant interviews.

Before participating in the study, written informed consent was obtained from the key informants, ensuring their voluntary involvement, and understanding of the study objectives. To uphold confidentiality, all collected data was securely stored on password-protected computers, with access restricted solely to the study team.

## Results

### Characteristics of Key informants

Of the 20 key informants, 13 were females. In terms of work duration, 9 had worked with PLHIV for 11–15 years and 9 were nurses. This is presented in Table 1.

The themes that emerged from our analysis comprise the commonly reported and managed NCDs, health system challenges associated with managing both NCDs and HIV, justifications advocating for integrated management, and recommendations for achieving successful integration. Table 2 provides a detailed presentation of all identified themes and their corresponding subthemes.

### The commonly reported and managed NCDs among people living with HIV

Metabolic and cardiovascular diseases especially diabetes and hypertension were reported to be common among PLHIV.

**Table 1. Characteristics of key informants.**

| KI# | SEX | CADRE | PERIOD WORKING WITH PLHIV (Years) |
|-----|-----|-------|-----------------------------------|
| 01 | M | Counselor | 14 |
| 02 | F | Nurse | 15 |
| 03 | F | Nurse | 6 |
| 04 | M | Doctor | 1 |
| 05 | F | Nurse | 12 |
| 06 | F | Nurse | 14 |
| 07 | F | Clinical Officer | 6 |
| 08 | M | Clinical Officer | 7 |
| 09 | M | Clinical Officer | 8 |
| 10 | F | Nurse | 13 |
| 11 | F | Nurse | 5 |
| 12 | F | Clinical officer | 9 |
| 13 | M | Public Health | 5 |
| 14 | F | Public Health | 7 |
| 15 | F | Nurse | 11 |
| 16 | F | Nurse | 10 |
| 17 | F | Clinical officer | 11 |
| 18 | F | Nurse | 5 |
| 19 | M | Counselor | 14 |
| 20 | M | Doctor | 11 |

*"Metabolic conditions are commonly reported among PLWHIV on care. . . . . . Diabetes is less common as compared to hypertension. . . . . .these conditions are common among PLWHIV compared to those who are HIV negative".***(KI09).**

*"When you take blood pressure say for 10 people, two to three will have high blood pressure"* **(KI 02).**

*"During counseling sessions, a lot of clients express concerns that they are experiencing blood pressure like symptoms. . . .when their blood pressures are measured, most are told that they are hypertensive. . . ."***(KI 19).**

Mental health conditions were also reported to be common among PLHIV at Kalisizo Hospital, prominent among others was depression.

*"In addition to other mental health conditions, depression is common among our patients. . . .. One of our client was severely depressed after being diagnosed with diabetes and hypertension. . . .. she attempted suicide twice, though we were able to offer her psychotherapy and thank God, she's still alive. . . . . ..many more have depression"* **(KI 11).**

*"Depression is like a sister to HIV at the hospital. . . .Many will become worried to the extent of failing to perform normal duties and on investigation, a diagnosis comes positive for depression"* **(KI 20).**

*"Depression is very common here. . .one in every three clients will experience depressive symptoms at least one a year"* **(KI 01).**

*"Generally, mental health concerns are highly prevalent here. . .We even have clients who have ever attempted suicide. . ."* **(KI 15).**

**Table 2. Themes and subthemes.**

| Theme | Sub-themes | Number (%) |
|---|---|---|
| Commonly reported and managed NCDs | Depression | 18 (90%) |
| | Hypertension | 16 (80%) |
| | Diabetes | 10 (50%) |
| | Cancers | 4 (20%) |
| | Kidney diseases | 2 (10%) |
| | Other NCDs | 1 (5%) |
| Health systems challenges in managing NCDs among PLHIV | Polypharmacy | 16 (80) |
| | Poor communication from specialists to lower cadre staff. | 16 (80) |
| | Inadequate resources | 12 (60) |
| | Unsupportive clinical guidelines | 10 (50) |
| | Limited knowledge | 8 (40) |
| Justification for integrated management. | Reduced cost | 19 (95) |
| | Reduced time | 18 (90) |
| | Reduced workload | 17 (85) |
| | Care for patients. | 10 (50) |
| | Improvement in treatment adherence | 10 (50) |
| Recommendations addressing hindrances to integrated management. | Training | 20 (100) |
| | Resource allocation | 18 (90) |
| | Time allocation | 16 (80) |
| | Involvement of relevant stakeholders in integrated management planning and implementation. | 13 (65) |

## Health systems challenges in managing HIV patients with NCDs

**Polypharmacy.** Although health workers perceived that most of their patients had low viremia, they noted challenges in coordinating medications, managing adverse events, handling unwanted drug interactions when coordinating HIV management with treatments for NCDs. Dealing with multiple medications and medication interactions were common concerns of health workers:

> *"People living with HIV who have NCDs come to the hospital feeling sad due to multiple medications. . . Some will even tell you they are done with certain drugs and others say they are tired of taking all drugs. . .we sometimes become confused how well to handle interaction given that you prescribe these drugs knowing possibilities of interaction and adverse events".* **(KI 08)**.

> *"A patient will come with drug related complaints, and you may end up giving them all your attention and ignoring your other patients. . ..managing drug interaction and adverse event is burdensome. . . . . . . . . . .".* **(KI 02).**

> *"Polypharmacy has always challenged us. . ...I don't know if drug manufacturers could think of supplying drugs that can treat multiple non-communicable diseases. . .this could help to reduce our major hurdle: Polypharmacy"* **(KI 20).**

**Poor communication and limited support from highly specialized physician.** Health workers reported grappling with challenges related to referrals to higher level healthcare providers who often failed to maintain effective communication and provide timely feedback. The

process of referring clients to specialists was hindered by delays in receiving information about their availability; specialists sometimes offered appointments without adequate communication, and in some instances, failed to notify the unavailability to keep the scheduled appointment. This inconsistency in communication created dissatisfaction among patients during such occurrences. One KI said,

*". . ..The NCD clinic typically operates on Thursdays and Fridays; however, there are instances where we experience extended periods, sometimes a week or more, without any consultative visits. Unfortunately, there is inconsistency in the scheduling, with visits occasionally occurring when our clients are not adequately informed, resulting in missed consultations for our clients. . ."* **(KI 18).**

Another communication concern was the failure to document reports from these consultations, leading to significant delays in seeking care by the clients. One KI said,

*"At times, our specialist providers send clients for prescriptions and lower-level management without providing essential documentation. This absence of proper documentation poses a challenge in delivering timely and accurate treatment to our patients. . . . . . . .."* **(KI 04).**

*"We sometimes refer patients to specialized physicians who sometimes don't communicate effectively with us. They'll keep quiet and only communicate back to us occasionally".* **(KI 06).**

*"The specialists communicate to us occasionally whenever we refer patients, and we are sometimes forced to wait for their communication before offering care and treatment to these patients and this makes the treatment process difficult. . .."* **(KI 09).**

**Inadequate resources to manage NCDs.**   Inadequate resources also challenge NCD management among PLHIV at the Hospital. Managing NCDs is difficult for practitioners especially when compounded with socioeconomic deprivation and limited budgetary allocation. Managing NCDs is financially costly yet the budget allocated to care, and treatment of these conditions is limited. One KI noted:

*"Some of these conditions especially hypertension and type 2 diabetes may require primary and secondary care, outpatient visits, and hospital admissions and these strain the Hospital's limited resources".* **(KI 20).**

Another KI recounted:

*"Each of these conditions is expensive to manage independently and you can imagine if an individual has multiple conditions".* **(KI 04).**

Another KI stated that:

*"When compared to their HIV negative counterparts, PLHIV suffer from condition which weaken their immunity and makes them vulnerable to more infections. Some of these conditions require out-of-pocket expenditure, yet most of these persons are in dire economic state making it difficult to buy medicines that they require to treat these conditions. . . . . .they are required to feed well to stay healthy. However, some of these individuals do not have money to buy food required to maintain healthy diet prior to medication and they sometimes come to*

*the facility seeking some support and we are sometimes required to give financial and non-financial support in terms of food making it difficult for us and the patients".* **(KI 09).**

**Unsupportive clinical guidelines.** The clinical guidelines focus on management of single conditions, and this hinders NCD management among PLHIV. One KI said:

*"It may be easier for us to manage HIV alone. However, if physical and mental health conditions are present, we are unable to consult the clinical guidelines on the proper management of these conditions and this affects our work . . ..".* **(KI 07).**

Another KI said:

*". . .You can only consult the available clinical guidelines if your patient has a single condition. . .when managing multiple conditions, the available clinical guidelines become less supportive. . ."* **(KI 13).**

**Limited knowledge and training in NCD management.** Lack of specialized knowledge, training and expertise also hinders NCD management at the Kalisizo Hospital. One of the KIs said:

*"Our training both in class and through Continuous medical education makes us competent in managing single conditions. Managing NCDs alongside HIV is challenging to most of us because it is not part of our training. . ..most of staff are clinical officers and nurses and are trained to manage single conditions. . . . . .even when we check our clinical guidelines, they focus on single conditions. . . . . .".* **(KI 12).**

*"A few of us here can manage multiple conditions simultaneously and for most, we feel we have a lot of knowledge gaps. . ..".* **(KI 18).**

## Perceptions of health workers regarding integrated management of NCDs in routine care

Out of the 20 key informants, a majority (fifteen) strongly advocated for the integrated management of NCDs in routine care. They substantiated their support by highlighting the crucial reasons for integration and offering insightful recommendations aimed at ensuring the success of integrated management.

*"Personally, I think by integrating non-communicable diseases in routine HIV care, it will reduce our workload. . ..we will have more time and we will be able to give our clients the kind of care they deserve, and this will improve non-communicable disease management. . ."* **(KI 20).**

*"By managing multiple conditions simultaneously, it saves time, costs and the burden on our side".* **(KI 17).**

*"Although it is important to integrate NCDs in routine care, a lot of efforts are desired including providing us with continuous medical education and training, increasing the resource base. . ..".* **(KI 20).**

Five key informants who did not recommend integrated management expressed concerns, highlighting issues such as insufficient budgetary allocation, the anticipated additional time required, and inadequate technical knowledge. For instance,

*"I don't think integration is what we need. Our budgetary allocation and human resource capacity cannot facilitate it and I don't think we are ready for it yet..."* **(KI 14).**

*"I am not knowledgeable enough about managing non-communicable diseases and I don't recommend integration"* **(KI 02).**

## Discussion

Our qualitative study explored the commonly managed NCDs among PLHIV by the health workers in South Central Uganda. The study also explored health system challenges associated with NCD management and investigated the perceptions of health workers regarding the integrated management of NCDs within routine HIV care at Kalisizo and Rakai Hospitals. Health workers commonly addressed hypertension and mental health conditions, particularly depression. However, they faced several challenges, including polypharmacy, insufficient communication from specialists to lower cadre health workers following referrals, limited financial and human resources, clinical guidelines that lacked support for managing NCDs in routine HIV care, and a deficiency in knowledge and skills needed to manage such conditions effectively.

In our study, health workers noted that a significant proportion of PLHIV with comorbid NCDs received treatment for conditions such as hypertension and mental health concerns, particularly depression. This observation aligns with the findings documented in previous studies [50, 51]. The consistent pattern observed may be attributed to the increased vulnerability of PLHIV to mental health challenges due to the self and public stigma associated with HIV [52]. Moreover, the role of HIV infection and ART toxicities in triggering chronic inflammation and immune activation within the body, which has the potential to disrupt blood vessel function may partly explain the high hypertension burden among PLHIV [53].

Additionally, health workers highlighted numerous challenges in managing HIV and NCDs. Notably, polypharmacy and the occurrence of adverse events were identified as common issues, consistent with findings in existing literature [54, 55]. The complexity of multiple drug therapy in the management of various diseases poses a significant challenge for both patients and physicians [56, 57]. Managing multiple diseases often becomes intricate as guidelines typically focus on individual diseases, offering guidance on when to initiate new medications but providing limited direction on when to discontinue them. This gap in guidance becomes especially apparent in the context of Long-Term Conditions (LTCs), where patients with a higher number of conditions are likely to be prescribed an increased number of medications [57]. Unfortunately, polypharmacy frequently leads to complications such as drug-disease interactions and drug-drug interactions [58].

The sub-theme of limited communication from higher-level health workers to lower-level health workers emerged prominently in our study, aligning with observations from prior research. Health workers noted that specialists sometimes did not keep scheduled appointments, and hardly communicated changes in their appointments with the clients, which affected consultative visits. Notably, a study on multimorbidity highlighted the challenges arising from the involvement of various medical specialists, each emphasizing the importance of their specific guidelines, coupled with poor communication from specialists and hospitals to family physicians, making coordination and medication management challenging. Despite these challenges, lower cadre workers expressed a desire for the input of specialists [58]. Our

study findings also underscored unsupportive clinical guidelines, insufficient resources, and a lack of knowledge and skills in integrating NCD management into routine HIV care as significant challenges in managing HIV patients with comorbid NCDs. These results resonate with findings from other studies [38, 39, 59].

Most of the health workers expressed a preference for the integrated management of Non-Communicable Diseases (NCDs) in routine care. Their preference was grounded in practical reasons for integration, including the potential to save time and financial resources. Furthermore, their support was fortified by insightful recommendations, notably emphasizing the importance of training health workers in integrated care. These findings align with the quantitative results reported in [60]. Moreover, a qualitative study conducted in Tanzania echoed similar sentiments, reporting that both health workers and clients favored the integrated management of HIV and NCDs in routine care [61].

## Study strength

This study boasts several strengths. Notably, it was carried out by proficient and experienced research assistants with advanced expertise in data collection and probing skills. This high level of competency undoubtedly played a crucial role in significantly enhancing the quality of the collected data. Furthermore, the study's multi-facility approach offered a more comprehensive understanding of the subject matter.

## Study limitations

This study has a notable limitation as it did not include interviews with District Health officials. To address this, we suggest that future studies incorporate interviews with District Health staff to provide a more comprehensive understanding of the subject matter and ensure a well-rounded assessment of NCD integration in routine care.

## Conclusions

The successful management of NCDs among PLHIV necessitates the integration of NCD management into routine HIV care. This integration, coupled with the development of supportive clinical guidelines, is essential for addressing the complex healthcare needs of individuals with comorbid conditions. Additionally, establishing timely, focused, and adequate communication from higher to lower-level health workers is imperative for enhancing coordination and ensuring comprehensive patient care. These measures collectively contribute to a more effective and holistic approach to healthcare for PLHIV, addressing both HIV and NCD challenges.

## Supporting information

**S1 File. Inclusivity in global research.** https://doi.org/10.6084/m9.figshare.25551813. (DOCX)

**S1 Checklist. Minimum dataset.** https://doi.org/10.6084/m9.figshare.25262521.v1. (DOCX)

## Acknowledgments

We would like to thank Kalisizo Hospital and Rakai Hospital for hosting this study. Additionally, we express our heartfelt appreciation to all the participants whose invaluable contributions made this research possible.

**Disclaimer:** The findings and conclusions in this report are those of the author(s) and do not necessarily represent the official position of the Rakai Health Sciences Program and African Medical and Behavioral Sciences Organization.

## Author Contributions

**Conceptualization:** Asani Kasango, Alex Daama.

**Data curation:** Asani Kasango, Alex Daama, Lilian Negesa.

**Formal analysis:** Asani Kasango, Alex Daama.

**Funding acquisition:** Asani Kasango, Alex Daama.

**Investigation:** Asani Kasango, Alex Daama.

**Methodology:** Asani Kasango, Alex Daama, Lilian Negesa.

**Project administration:** Asani Kasango.

**Resources:** Asani Kasango.

**Software:** Asani Kasango, Alex Daama.

**Supervision:** Asani Kasango, Alex Daama.

**Validation:** Asani Kasango, Alex Daama.

**Visualization:** Asani Kasango.

**Writing – original draft:** Asani Kasango, Alex Daama, Lilian Negesa.

**Writing – review & editing:** Asani Kasango, Alex Daama, Lilian Negesa.

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
