## [Decision Letter · Decision Letter 0]

30 Jan 2024

PONE-D-23-38985Challenges in managing Persons living with HIV with non-communicable diseases and the perception of health workers towards integrated management of non-communicable diseases during routine HIV care in South Central Uganda: A qualitative study.PLOS ONE

Dear Dr. Kasango,

Thank you for submitting your manuscript to PLOS ONE. After careful consideration, we feel that it has merit but does not fully meet PLOS ONE’s publication criteria as it currently stands. Therefore, we invite you to submit a revised version of the manuscript that addresses the points raised during the review process.

 Please submit your revised manuscript by Mar 15 2024 11:59PM. If you will need more time than this to complete your revisions, please reply to this message or contact the journal office at plosone@plos.org. Please include the following items when submitting your revised manuscript:A rebuttal letter that responds to each point raised by the academic editor and reviewer(s). You should upload this letter as a separate file labeled 'Response to Reviewers'.A marked-up copy of your manuscript that highlights changes made to the original version. You should upload this as a separate file labeled 'Revised Manuscript with Track Changes'.An unmarked version of your revised paper without tracked changes. You should upload this as a separate file labeled 'Manuscript'.If applicable, we recommend that you deposit your laboratory protocols in protocols.io to enhance the reproducibility of your results. Protocols.io assigns your protocol its own identifier (DOI) so that it can be cited independently in the future. For instructions see: https://journals.plos.org/plosone/s/submission-guidelines#loc-laboratory-protocols. Additionally, PLOS ONE offers an option for publishing peer-reviewed Lab Protocol articles, which describe protocols hosted on protocols.io. Read more information on sharing protocols at https://plos.org/protocols?utm_medium=editorial-email&utm_source=authorletters&utm_campaign=protocols.

We look forward to receiving your revised manuscript.

Kind regards,

Elizabeth S. Mayne, M.D.

Academic Editor

PLOS ONE

Journal Requirements:

3. In the online submission form, you indicated that "The datasets used and/or analyzed during the current study are available from the corresponding author on reasonable request."

**Additional Editor Comments:**

Both reviewers felt that the manuscript had value. Reviewer 2 was concerned that the introduction was not comprehensive enough and that the discussion required reorganisation. The reviewer also addressed the concern that the consent was in Luganda but the interviews were in English - the authors should please address this explicitly since this could introduce bias. The other reviewer had comments regarding the escalation of care and these should also be considered.

Reviewers' comments:

Reviewer's Responses to Questions

**Comments to the Author**

1. Is the manuscript technically sound, and do the data support the conclusions?

Reviewer #1: Yes

Reviewer #2: Partly

2. Has the statistical analysis been performed appropriately and rigorously? 

Reviewer #1: Yes

Reviewer #2: Yes

3. Have the authors made all data underlying the findings in their manuscript fully available?

Reviewer #1: Yes

Reviewer #2: Yes

4. Is the manuscript presented in an intelligible fashion and written in standard English?

Reviewer #1: Yes

Reviewer #2: Yes

5. Review Comments to the Author

Reviewer #1: The paper addressed important aspects of NCD integration into HIV mamangement silos that are as yet, unresolved. The topic is particularly relevant for LMICs - with far fewer resources at hand to address the complexities of health systems integration. Whie sounding relatively simple, integrated care is difficult - as shown by this study.

The paper is well written, and the authors are to be congratulated.

I have attached minor comments for review in the manuscript pdf.

Reviewer #2: The introduction is scanty and does not give adequate background. Enrichment with more literature references would be most welcome.

Reference the statistical software used to analyse eg. Nvivo software.

Table 1: Cadre: Participants 14-18 are written in caps, change to small caps for consistency.

Line 98 “ In terms of work duration, 11 had worked with PLHIV for 11-15 years and 9 were nurses” However, in Table 1: only 9 participants had worked for 11-15 years and 6 were nurses

There is no consistency with writing abbreviations. Eg: PLHIV vs PLWHIV

Under ethics statement they mentioned that the consents were administered in either Luganda or English, however, in line 85 they mentioned that interviews were conducted in English.

The statistical analysis is not in deep described and is not clear which statistical analysis was performed for descriptive analysis.

Under the method section, clarity on how the participant number was calculated to achieve satisfactory statistical power for appropriate interpretation of results is needed.

The discussion section should be deep reorganized. Discussion section should relates to the literature review and research questions, and making an argument in support of your overall conclusion.

6. PLOS authors have the option to publish the peer review history of their article (what does this mean?). If published, this will include your full peer review and any attached files.

Reviewer #1: **Yes: **June Fabian

Reviewer #2: No

---

## [Author Response · Author response to Decision Letter 0]

22 Feb 2024

POINT-BY-POINT REBUTTAL LETTER 

 We are grateful for the opportunity to revise our manuscript titled “Challenges in managing HIV and non-communicable diseases and health workers’ perception regarding integrated management of non-communicable diseases during routine HIV care in South Central Uganda: A qualitative study”. In response to the insightful feedback from the editor and reviewers, we have revised the manuscript, incorporating changes and additions where necessary. Below, we provide a detailed response to each comment raised by the reviewer. The format of our responses is organized as follows: a) Comments from editors or reviewers are presented in standard text. b) Our responses to each comment are indicated directly beneath the comment and are highlighted in red for clarity. 

Journal Requirements 

https://journals.plos.org/plosone/s/file?id=wjVg/PLOSOne_formatting_sample_main_body.pdf and https://journals.plos.org/plosone/s/file?id=ba62/PLOSOne_formatting_sample_title_authors_ affiliations.pdf 

We have checked the templates and made the necessary changes, including file naming. we have also formatted the manuscript to reflect the PLOS One formatting style.

All findings derived from the study are fully disclosed and published in the main manuscript body. 

3. In the online submission form, you indicated that "The datasets used and/or analyzed during the current study are available from the corresponding author on reasonable request."

This policy applies to all data except where public deposition would breach compliance with the protocol approved by your research ethics board. If your data cannot be made publicly available for ethical or legal reasons (e.g., public availability would compromise patient privacy), please explain your reasons on resubmission and your exemption request will be escalated for approval.. 

All findings derived from the study are fully disclosed and published in the main manuscript body. 

Your ethics statement should only appear in the Methods section of your manuscript. If your ethics statement is written in any section besides the Methods, please move it to the Methods section and delete it from any other section. Please ensure that your ethics statement is included in your manuscript, as the ethics statement entered into the online submission form will not be published alongside your manuscript.

The ethics statement has been moved the methods section of our manuscript. 

The current reference list is complete and correct. We removed retracted papers from the references.

- Reviewer #2: The introduction is scanty and does not give adequate background. Enrichment with more literature references would be most welcome.

We have revised our introduction and added more literature.

Reference the statistical software used to analyse eg. Nvivo software. - 

We have referenced the statistical software used (Nvivo).

Table 1: Cadre: Participants 14-18 are written in caps, change to small caps for consistency.

We have changed Table 1 Cadre: Participants 14-18 to small caps to ensure consistency.

 Line 98 “ In terms of work duration, 11 had worked with PLHIV for 11-15 years and 9 were nurses” However, in Table 1: only 9 participants had worked for 11-15 years and 6 were nurses.

 We have made the necessary changes as follows: Of the 20 key informants, 13 were females. In terms of work duration, 9 had worked with PLHIV for 11-15 years and 9 were nurses.

There is no consistency with writing abbreviations. Eg: PLHIV vs PLWHIV

We have made our abbreviations consistent and maintained PLHIV.

Under ethics statement they mentioned that the consents were administered in either Luganda or English, however, in line 85 they mentioned that interviews were conducted in English.

We administered our consent in English, and that was a typo.

The statistical analysis is not in deep described and is not clear which statistical analysis was performed for descriptive analysis.

We have revised the data analysis section to reflect statistical analysis.

Under the method section, clarity on how the participant number was calculated to achieve satisfactory statistical power for appropriate interpretation of results is needed.

We were guided by the saturation principle, and we reached saturation at the 20th interview.

The discussion section should be deep reorganized. Discussion section should relates to the literature review and research questions, and making an argument in support of your overall conclusion.

We have reorganized the discussion section and it now relate well with literature review, research questions and supports the overall conclusion.

Does this refer to efforts to escalate care for those in need - which doesnt come out clearly in this paper. It would help to clarify 1. could health care workers escalate their patients as needed into advanced tiers of care 2. was the poor health communication around efforts to escalate care or feedback from practitioners to whom care was escalated

This was addressed in the manuscript body.

---

## [Decision Letter · Decision Letter 1]

2 Apr 2024

Challenges in managing HIV and non-communicable diseases and health workers’ perception regarding integrated management of non-communicable diseases during routine HIV  care in South Central Uganda: A qualitative study.

PONE-D-23-38985R1

Dear Dr. Kasango,

We’re pleased to inform you that your manuscript has been judged scientifically suitable for publication and will be formally accepted for publication once it meets all outstanding technical requirements.

Kind regards,

Elizabeth S. Mayne, M.D.

Academic Editor

PLOS ONE

Additional Editor Comments (optional):

Reviewers' comments:

Reviewer's Responses to Questions

**Comments to the Author**

1. If the authors have adequately addressed your comments raised in a previous round of review and you feel that this manuscript is now acceptable for publication, you may indicate that here to bypass the “Comments to the Author” section, enter your conflict of interest statement in the “Confidential to Editor” section, and submit your "Accept" recommendation.

Reviewer #2: All comments have been addressed

2. Is the manuscript technically sound, and do the data support the conclusions?

Reviewer #2: Yes

3. Has the statistical analysis been performed appropriately and rigorously? 

Reviewer #2: Yes

4. Have the authors made all data underlying the findings in their manuscript fully available?

Reviewer #2: Yes

5. Is the manuscript presented in an intelligible fashion and written in standard English?

Reviewer #2: Yes

6. Review Comments to the Author

Reviewer #2: The authors have addressed all the minor comments requested to my satisfaction. It is acceptable for publication.

7. PLOS authors have the option to publish the peer review history of their article (what does this mean?). If published, this will include your full peer review and any attached files.

Reviewer #2: No

---

## [Editor Report · Acceptance letter]

10 Aug 2024

PONE-D-23-38985R1 

PLOS ONE

Dear Dr. Kasango, 

I'm pleased to inform you that your manuscript has been deemed suitable for publication in PLOS ONE. Congratulations! Your manuscript is now being handed over to our production team.

Kind regards, 

on behalf of

Dr. Elizabeth S. Mayne 

Academic Editor

PLOS ONE